# DaylilyNet: A Multi-Task Learning Method for Daylily Leaf Disease Detection

**DOI:** 10.3390/s23187879

**Published:** 2023-09-14

**Authors:** Zishen Song, Dong Wang, Lizhong Xiao, Yongjian Zhu, Guogang Cao, Yuli Wang

**Affiliations:** 1Shanghai Institute of Technology, College of Computer Science and Information Engineering, Shanghai 201418, China; helloszs@foxmail.com (Z.S.); lyexp@163.com (L.X.); guogangcao@163.com (G.C.); wangyuli1252@gmail.com (Y.W.); 2Ningbo Minjie Information Technology Co., Ltd., Ningbo 315040, China

**Keywords:** daylily disease detection, complex background interference, multi-task learning

## Abstract

Timely detection and management of daylily diseases are crucial to prevent yield reduction. However, detection models often struggle with handling the interference of complex backgrounds, leading to low accuracy, especially in detecting small targets. To address this problem, we propose DaylilyNet, an object detection algorithm that uses multi-task learning to optimize the detection process. By incorporating a semantic segmentation loss function, the model focuses its attention on diseased leaf regions, while a spatial global feature extractor enhances interactions between leaf and background areas. Additionally, a feature alignment module improves localization accuracy by mitigating feature misalignment. To investigate the impact of information loss on model detection performance, we created two datasets. One dataset, referred to as the ‘sliding window dataset’, was obtained by splitting the original-resolution images using a sliding window. The other dataset, known as the ‘non-sliding window dataset’, was obtained by downsampling the images. Experimental results in the ‘sliding window dataset’ and the ‘non-sliding window dataset’ demonstrate that DaylilyNet outperforms YOLOv5-L in mAP@0.5 by 5.2% and 4.0%, while reducing parameters and time cost. Compared to other models, our model maintains an advantage even in scenarios where there is missing information in the training dataset.

## 1. Introduction

Daylilies are extensively cultivated in China and serve as a significant agricultural crop, providing an economic lifeline to rural communities. Recently, the government has been promoting the daylily farming industry as a means to alleviate rural poverty and expand cultivated land. However, daylilies are susceptible to diseases, especially during the early to mid-growth stages, which can result in reduced income. Given the highly contagious nature of daylily diseases, timely detection is imperative. Currently, crops mainly rely on manual inspection, which is time-consuming.

In recent years, propelled by advancements in computer vision, image recognition techniques based on deep learning have found applications in various fields, including facial recognition, pedestrian detection, autonomous driving, object recognition, and object tracking [1,2,3]. Deep learning methods for object detection involve locating the target region and performing category classification. Due to their outstanding performance, researchers in the field of plant disease detection have shifted their focus from traditional visual inspection methods to deep learning approaches.


**Types of object detection algorithms in deep learning**


Presently, object detection models in deep learning can be categorized into single-stage and two-stage detection models. Furthermore, the classification distinguishes them based on using anchor boxes into anchor-free and anchor-based approaches. Remarkable anchor-free models include FCOS [4] and CenterNet [5].

Object detection using these models encompasses two stages: object localization and object classification. Theoretically, two-stage networks offer superior object localization performance, while single-stage networks provide faster detection speed and lower memory consumption. This study proposes a swifter, more accurate object detection algorithm based on the single-stage approach to balance detection speed and performance.

The principal distinction between deep-learning-based approaches and traditional machine learning methods resides in their fundamental disparities. Deep learning algorithms harness substantial volumes of data and employ gradient descent algorithms to iteratively refine the parameters of feature extractors, thereby adapting them without necessitating manual feature engineering. This trait empowers deep learning models to demonstrate enhanced data-fitting capabilities. Furthermore, deep learning models comprise deeper layers, enabling them to acquire and extract intricate semantic features from the data. As a result, deep learning methods often surpass manually designed feature extractors. Particularly noteworthy is the fact that deep-learning-based techniques have led to significant progress in plant disease detection in recent years.


**Development of plant disease detection technology based on deep learning**


Li et al. [6] employed the Feature Pyramid Network (FPN) along with the Precise Region of Interest (PROI) pooling module to enhance the detection accuracy of Faster R-CNN for small targets with complex backgrounds. Xiong et al. [7] introduced a novel image segmentation algorithm called AISA, effectively eliminating background information. Their system achieved an identification accuracy of over 80% for 27 diseases across six different crops in both laboratory and field environments. Chen et al. [8] developed a comprehensive framework for extracting pest and disease features, localizing diseases, and performing classification. They utilized a sliding window detection algorithm. Zhou et al. [9] presented a method for detecting rice diseases using the Faster R-CNN framework, achieving an average detection accuracy of 95% and a processing speed of 0.66 s per image. Ghoury et al. [10] employed transfer learning methods to compare the performance of SingleSSD_MobileNetv1 and Faster R-CNN in discerning healthy and diseased grape leaves. Their investigation revealed that SingleSSD_MobileNetv1 boasted faster detection speeds, while Faster R-CNN demonstrated superior detection accuracy. Fuentes et al. [11] utilized three models, namely Faster R-CNN, R-FCN, and SSD, for accurately localizing and classifying regions affected by tomato leaf diseases. Among these models, ResNet50 served as the feature extractor, attaining an average precision (mAP) of 85.98%. Subsequently, Fuentes et al. [12] introduced enhancements to the Faster R-CNN model by incorporating secondary classification units to improve the identification of secondary categories and reduce false detection rates, resulting in a 13% increase in mAP. Bhatt et al. [13] introduced an innovative technique for detecting pests and diseases in tea gardens under uncontrolled conditions. They employed YOLOv3 as the object detection model and achieved a mAP@0.5 performance of 86% with an IOU threshold of 50%. Singh et al. [14] established the PlantDoc dataset for detecting plant diseases, emphasizing the detection speed for embedded devices. Their research utilized MobileNet and SSD as detection models.


**The state-of-play and direction of research**


(1) Considering further model enhancement using two-stage detection approaches to improve detection accuracy in complex scenes.

(2) Refining single-stage models to decrease parameters and computations while retaining detection accuracy for resource-constrained embedded devices. However, these directions often emphasize complex scenes, where a single diseased leaf is positioned at the image’s center with a blurred “background” during the capturing process, diverging from real-world scenarios.

(3) These approaches predominantly adopt the CNN-based approach, overlooking global contextual information. These approaches may not always lead to enhancements in accuracy when implemented for practical scene detection.


**Limitations with existing plant disease detection based on deep learning**


(1) Much research has focused on target objects with less complex or noise-free backgrounds. Nevertheless, the efficacy of these methods tends to be subpar in real-world detection scenarios.

(2) The establishment of the model’s global feature extraction capability is often inadequate. Most plant disease detection models rely on convolutional neural networks (CNNs). While CNN-based segmentation methods have demonstrated improved outcomes in coarse-grained object detection, these approaches confine the model’s focus to a limited neighborhood, thereby diminishing its capacity to distinguish small disease targets from intricate backgrounds and foliage.


**Research progress in multi-task learning**


In recent years, many multi-task learning models have been proposed to utilize a single backbone network to tackle two or more distinct tasks. The study conducted by YOLOP [15] noted the existence of correlations among tasks and connections between the objects involved in these tasks when performing various categories of tasks. Through the establishment of shared weight parameters within a multi-task learning framework, the tasks’ performance can be enhanced. Araki et al. [16] integrated a multi-task learning strategy into the visual detection sub-model of a robot’s object grasping system, effectively improving the detection accuracy for both object detection and semantic segmentation tasks. Chen et al. [17] introduced the UNet as a feature extraction task and utilized its Semantic Segmentation Head to aid the training process for infrared small-target detection. This approach led to enhanced average precision and an increased detection rate of small targets.


**Research progress in small object detection**


It is imperative to address the challenge of detecting small objects. When dealing with small object detection, the scarcity of target features presents a formidable challenge for conventional CNN structures to process these feature points adeptly. Despite the potential alleviation of this issue through introducing FPN, the detection performance of small objects might still be compromised in the presence of complex backgrounds. Sun et al. [18] devised a variant of detection transformer (DETR) composed exclusively of an encoder to tackle the challenge of feature extraction from small objects. This design improves the detection accuracy of small objects, possibly reducing detection precision for larger objects. Xu et al. [19] introduced a dual-keyword Transformer architecture to enhance the efficacy of small object detection further. The incorporation of the Transformer was observed to mitigate the drop in detection accuracy when small objects become occluded. Dubey et al. [20] introduced the improving small objects detection using Transformer (SOF-DETR) method, which utilizes a normalized inductive bias for object detection. Using a self-attention mechanism to capture spatial relationships between objects situated at varying distances within an image enhances the efficacy of small object detection.


**Analysis and solutions**


Given the context above, we can surmise that the detection of daylily diseases in real-world scenarios is challenged by background interference complexities and substantial variations in target scales. These challenges have resulted in the suboptimal performance of most deep-learning methods in this field. In an endeavor to tackle these challenges, taking inspiration from YOLOP, Transformer [21], and CvT [22], this study presents a pioneering daylily disease detection methodology named DaylilyNet. The introduced approach amalgamates multi-task learning with a separable attention mechanism.

To counteract CNN models’ insufficient global feature correlation capabilities, we incorporate the Separable Attention module derived from MobileViTv3 [23]. Furthermore, while tackling the problem of feature displacement inherent in the conventional Feature Pyramid Network (FPN) architecture, we introduce the Feature-aligned Pyramid Network (FaPN) and optimize it to amplify feature fusion capabilities and enhance flexibility.

Moreover, we devise a Decoupled Head to segregate the target classification and localization tasks in object detection. We introduce a disease leaf Segmentation Head alongside its corresponding loss function to mitigate the challenge posed by misclassifying backgrounds. This joint optimization enhances the model and reduces the likelihood of incorrectly classifying backgrounds as disease targets. The experimental findings demonstrate the successful enhancement of the model’s proficiency in detecting daylily leaf diseases through these improvements.


**Organizational structure of the article**


Section 2 introduces the methodology proposed in this study. Section 3 provides information about the dataset and presents the evaluation metrics. Section 4 presents the experimental results and analysis. Section 5 discusses the experimental findings. Section 6 concludes the content and findings of this paper.

## 2. Materials and Methods

YOLOv5 achieves object localization and classification by directly regressing candidate box positions and categories. It provides a straightforward and convenient approach while retaining reasonable accuracy. Considering the intricate backgrounds in real-world daylily disease images, scale variations from different distances to the camera for the same disease, and significant non-rigid disease shape disparities, we introduce DaylilyNet, built upon the foundation of YOLOv5. This approach harnesses the power of multi-task learning and a separable attention mechanism to tackle these challenges encountered in daylily disease detection.

### 2.1. The Architecture of DaylilyNet

DaylilyNet consists of an enhanced MobileViTv3 backbone (IMB), PAN structure with the improved Feature Adaptive Pyramid Network (IFPN), a leaf disease segmentation module (Seg), and the detection module named the Decoupled Head (DH), as depicted in Figure 1.

The FPN structure amplifies the semantic information from the original image within deep feature maps. The fusion of multi-scale feature information and the addition of semantic details contribute to the enhancement of detecting disease targets of smaller sizes. In this pathway, the feature maps at each scale are linked with the corresponding scale outputs from the bottom-up path, thereby expanding semantic information.

Drawing inspiration from the Inception structure [24] and to further enlarge the receptive field, the bottom-up pathway encompasses the simultaneous processing of inputs through a 5 × 5 and a 3 × 3 convolution module, followed by an element-wise summation of the resulting features from both modules. 

Additionally, IFPN is utilized to rectify the feature misalignment induced by upsampling and downsampling, thus augmenting the localization accuracy of the detection module. 

Finally, the features of disease-infected leaves are extracted through the Segmentation Head (Seg). The Segmentation Head employs the P2 feature map for multiple upsampling and feature extraction iterations, ultimately augmenting the spatial resolution of the feature maps to align with that of the original image. Furthermore, the disease target detection module comprises a Decoupled Head (DH).

### 2.2. Improved MobileViTv3 Backbone

In the context of complex-background object detection, the performance of the feature extraction layer holds paramount significance. Considering the extensive use of Transformers in medical image segmentation, the powerful global feature extraction capability of Transformers has been utilized to tackle the issue of diminished segmentation precision caused by the complex spatial relationships of objects [25]. Considering the application context and the resemblances in the features of the detection targets, this study has opted to replace a pure CNN structure with the Transformer architecture for feature extraction. However, owing to the presence of inductive bias, which results in inadequate detection accuracy when identifying small objects in images with complex backgrounds [26], a local feature extraction structure has been incorporated into the models. To mitigate the computational complexity of self-attention, the Separable Attention mechanism from MobileViTv3 [23], known for its linear complexity, has been adopted to capture the global spatial relations.

The architecture of the Improved MobileViTv3 backbone (IMB) is depicted in Figure 2. The process of obtaining feature maps C1 to C4 is denoted as the Embedding process. The Conv module is utilized for patch embedding, representing an improvement over the traditional patch embedding module employed in ViT. Unlike the latter, our method solely depends on convolutional modules for downsampling and modeling spatial relationships, reducing computational costs and parameter quantities [27].

The feature map C4 is fed into the MobileViTv3 (MV) module, which includes the LinearTransformer, a Transformer with linear computational complexity. This module facilitates global feature extraction and the correlation of features. Notably, due to the considerable semantic information contained within the input feature map, which possesses a spatial resolution of 40 × 40, this feature map is fed into the MV module to extract global spatial features. The final step entails fusing elements across different receptive fields using the Spatial Pyramid Pooling Fast (SPPF) module. However, the IMB employs a method that encompasses two pixels for feature correlation, resulting in an indirect reduction in the semantic resolution of the feature map and a weakening of the ability to extract features from small targets. This study addressed this aspect by employing full-pixel sampling and allocating weighted values to all spatial pixels.

The enhanced MV module, illustrated in Figure 3, consists of a global representation (GR) module, a local representation (LR) module, and a fusion module. The GR module establishes correlations within the global spatial context, while the LR module extracts features from local spatial regions. The integration of the LR module enhances the model’s ability to infer insights from two-dimensional spatial features in images, thereby expediting the model’s convergence. 

The structure of the linear transformer is depicted in Figure 4. We employ the Pre-norm approach [28] to improve the model’s convergence rate. In the original MobileViTv3 framework, a 1 × 1 convolutional layer was utilized in the FeedForward stage to facilitate interaction between features from different channels within the feature map. However, given the relatively small scale of plant leaf diseases, fully utilizing information from all channels within the feature map is crucial. Therefore, we utilize the Linear module to facilitate interaction among pixels at the same positions across all channels. This approach improves the accuracy of localizing small targets.

In MobileViTv3, the linear transformer utilizes the Separable self-attention (SSA) structure, as illustrated in Figure 5. When compared to the traditional Multi-Head Self-Attention (MHSA) mechanism, the linear transformer demonstrates lower computational complexity. The IMB integrates the separable self-attention mechanism SSA to mitigate computational complexity and reduce the number of parameters. The computational complexity of MHSA and SSA is presented in Table 1, where k represents the number of tokens, equivalent to the number of pixels involved in global spatial feature modeling.

When configuring SSA parameters, a patch size of 2 is employed. It divides the feature map into groups of four adjacent pixels per spatial domain, starting from the first pixel. For a feature map with a resolution of *h* × *w*, this results in (*h*/2) × (*w*/2) groups. Tokens in the same position within each group, referred to as tokens of the same color, are ordered using the unfold operation. The sequence length becomes (*h*/2) × (*w*/2). Applying this operation to the group feature map transitions the data shape from (*b*, *c*, *h*, *w*) to (*b*, *c,* 2 × 2, (*h*/2) × (*w*/2)). Then, the feature map is grouped along the channel dimension, creating a data shape of (*b*, *c*, 2 × 2, (*h*/2) × (*w*/2)). A 1 × 1 convolution is conducted with a channel number of 1 + 2 × *e*, where *e* represents the embedded dimension (set to 512 in our network). The feature map is divided into *I, K* (also known as *Q*), and *V*, each with 1, *e*, and *e* channel numbers, respectively. A Softmax operation is applied to the feature map *I* along the row dimension, resulting in a context scores matrix representing token contextual scores. This matrix is element-wise multiplied with the feature map *K*, producing a context vector denoted as $C_{V}$. The context vector encodes global information and is expanded along the row dimension with uniform values. This expansion results in a context matrix, which is element-wise multiplied with the feature map *V* to generate the global attention feature map. Finally, a 1 × 1 convolution layer with a channel number of e is applied to the feature map, followed by a Fold operation to restore its spatial feature map shape to (*b*, *c*, *h*, *w*).

In the end, the separable attention can be represented by the following Equation:(1)$$y=\left(\sum \left(\sigma (XW_{I})\times XW_{K}\right)\times ReLU(XW_{V})\right)W_{O}=C_{v}\times ReLU(XW_{V})\times W_{O},$$

In this equation, $X\in R^{d\times d}$ represents the input feature map, σ signifies the softmax function, and $W_{I}$, $W_{K}$, $W_{V}\in R^{d\times d}$ denote the weights of the convolutional layers that encode the input feature map *X* into *I*, *K*, and *V*. $W_{O}\in R^{d\times d}$ represents the weights of the 1 × 1 convolutional linear layer before the output. The variable d corresponds to the token dimension.

The context vector $C_{V}$ can be represented as:(2)$$C_{v}=\sum_{i=0}^{k}C_{s}(i)X_{K}(i)=\sum \left(\sigma (XW_{I})\times XW_{K}\right),$$

The context score $C_{S}$ can be represented as:(3)$$C_{s}=\sigma (XW_{I}),$$

### 2.3. PAN Structure with the IFPN

Compared to the feature pyramid network, the Path Aggregation Network (PAN) offers advantages in enhancing object detection capabilities. However, when employing traditional channel concatenation methods on upsampled and downsampled feature maps, feature misalignment issues arise, which can lead to a decrease in object localization accuracy. To tackle these concerns regarding feature misalignment, the proposed FaPN [29] is introduced to align feature maps. 

Moreover, we have observed that the existing fusion process involves a simple element-wise addition of two sets of feature maps, which fails to address the uneven contributions of these feature maps to the information. Consequently, we adapt the fusion mechanism from element-wise summation to element-wise weighted summation, thereby enhancing flexibility in the fusion process. This adjustment enables the network to assign varying importance to different sets of feature maps, thus bolstering the model’s robustness. The revised FaPN module is referred to as the Improved FaPN (IFPN). The overall workflow of IFPN is illustrated in Figure 6. The horizontally concatenated features undergo feature selection via FSM, and subsequently, the output feature map enters FAM to align the bottom-up feature maps. The resultant feature maps from FSM and FAM are combined through element-wise weighted summation using dynamically determined weights *W1* and *W2* during the network training process.

#### 2.3.1. FSM Module

In the original FPN framework, the fusion of bottom-up and lateral connection feature maps was achieved solely through simple concatenation, inadvertently leading to the incorporation of a substantial amount of redundant information from the lateral connection feature maps.

The introduction of the Feature Selection Module (FSM) aimed to address this issue. The structure of the FSM module is depicted in Figure 7. Within the branch responsible for channel feature importance scoring, the input features (input1) are initially subjected to an average pooling layer to compress spatial information. Subsequently, the output is directed to a 1 × 1 convolution module, which applies weighted scoring to the compressed information of each channel. The resulting scores are then nonlinearized through a sigmoid function, resulting in score1. This score1 is subsequently utilized to weigh the channel scores of each individual input1, consequently generating feature2. Feature2 is then combined with input1 through a residual connection and, ultimately, processed via a 1 × 1 convolutional layer to selectively retain channels, thereby effectively suppressing redundant features. 

#### 2.3.2. FAM Module

Feature Alignment Module (FAM) is introduced to address the issue of misalignment between the interpolated upsampled features from the shallow top-down layers and the bottom-up feature maps, which are downsampled using convolutional modules in the original FPN. The fusion of these two types of feature maps through element-wise addition or channel concatenation followed by a 1 × 1 convolution can adversely affect the performance of boundary box regression and classification in the prediction heads. Therefore, the feature fusion methods that rely on element-wise addition or channel concatenation can affect the prediction of object boundaries and result in misclassification during the prediction process. To overcome these problems, the FAM module is introduced to align the spatial positions of bottom-up feature maps. This method adjusts the positional offsets of convolutional kernel sampling during the feature map sampling process to map the features to their correct positions, thus achieving feature alignment. 

The workflow of the Feature Alignment Module is illustrated in Figure 8. The module takes two inputs: *L1*, which is the horizontally connected feature map obtained from the FSM output, and *T1*, which is the top-down feature map. Both feature maps have the same spatial resolution and channel number. Firstly, the two feature maps are concatenated along the channel dimensions. Then, a 1 × 1 convolutional layer with *C* channels is applied to extract the feature map *O1*, which represents the spatial pixel position offsets of the two feature maps. Subsequently, both *T1* and *O1* are fed into the deformable convolution module, where the feature offsets provided by *O1* guide the deformable convolution kernel to convolve *T1*, aligning the features based on the calculated offsets.

### 2.4. Decoupled Head

Due to the distinct requirements of the localization and classification tasks, using a single convolutional module for both tasks within a Non-Decoupled Head cannot effectively extract the features necessary for each task [30]. Consequently, in our proposed model, a Decoupled Head (DH) is employed for the detection of leaf disease targets. The DH module comprises three separate heads, as depicted in Figure 9. The feature maps of sizes 256 × 20 × 20, 256 × 40 × 40, and 256 × 80 × 80 are fed into DH modules 1, 2, and 3, respectively, to detect large, medium, and small targets in the image. The structures of the three DH modules are uniform, each consisting of four 1 × 1 convolutional modules and one 3 × 3 convolutional module.

After the feature maps are fed into the Decoupled Head, they undergo channel-wise fusion through a 1 × 1 convolutional layer with 256 channels, followed by a 3 × 3 convolutional layer with 256 channels to extract semantic information from neighboring pixels. This process enhances the localization capability by capturing local semantic information. The resulting feature maps are then simultaneously input into the Object (Obj.) branch, Classification (Cls.) branch, and Regression (Reg.) branch for box prediction, classification, and size estimation, respectively. These branches are constructed using 1 × 1 convolutional modules with channel numbers of 13, *cls**3, and 4*3, respectively. Here, 3 signifies the three aspect ratios of anchor boxes within each prediction grid. In the Obj. branch, 1 indicates the presence of a disease target in that grid, while *cls* represents the number of classes. In this study, there are three disease classes; hence, *cls* is set to 3. The Reg. branch is responsible for box localization, predicting the coordinates (*x*, *y*) and dimensions (*h*, *w*) of the bounding boxes.

### 2.5. Segmentation Head

Given that the model frequently faces challenges like occlusion between leaves and variable spatial distances between diseased leaves and the camera, it encounters variations in the scale or shape of the same type of disease, leading to false detections. Furthermore, the intricate background might cause the model to misinterpret background objects as disease objects. Comparatively, detecting or segmenting diseased leaves themselves is relatively easier and more accurate, as the scale of diseased leaves is generally larger than that of disease targets. Therefore, our proposed model incorporates a Segmentation Head for diseased leaves and introduces corresponding loss functions during training to jointly optimize the model, aiming to diminishing false detections.

Drawing inspiration from multi-task models such as YOLOP, we integrate a Segmentation Head (Seg) into the shallow layers of the model, given that segmenting diseased leaves necessitates contour and texture information. Sufficient information for the segmentation task can influence the parameter adjustments in the shallow feature extraction layers, indirectly impacting the deeper disease detection module and optimizing the parameter update direction to mitigate false detections in complex scenarios.

In our study, the semantic segmentation task involves only two classes: “diseased leaves” and “background”, rendering it relatively straightforward. Thus, we employ narrow-channel convolutional layers and the C3 module for feature extraction within the Segmentation Head. The structure of the Segmentation Head is depicted in Figure 10. It takes the input features and applies a 3 × 3 convolution followed by upsampling. The C3 module is subsequently employed to extract local spatial interaction information. This process is reiterated until the feature map is upsampled to match the dimensions of the input image. Finally, a 1 × 1 convolution is utilized to classify the superpixels, with a channel size of 2 to accommodate the two classes. Following this step, the module can classify each pixel in the original image into its respective category.

### 2.6. Loss Function

The workflow of single-stage object detection can be summarized as follows: regression for predicting the position and size of the target box, regression for estimating the confidence of target presence at that position, and classification of the target. In this study, the object detection task encompasses three distinct loss functions, corresponding to the tasks of box position regression, probability estimation of target presence at each location, and target classification. These loss functions are denoted as $L_{CIOU}$, $L_{conf}$ and $L_{seg}$.

The introduced CIoU (Complete Intersection over Union) builds upon DIoU [31] by introducing additional losses for scale, width, and height. This refinement aids in making the predicted boxes better aligned with the ground truth boxes.

Furthermore, this study introduces an additional task of semantic segmentation, aimed at segmenting the regions of diseased leaves within the image and guiding the optimization direction of the object detection task. With the incorporation of this task, a semantic segmentation loss function is formulated for optimization.

Consequently, the complete form of the loss functions can be represented as follows:(4)$$L_{total}=L_{CIOU}+L_{conf}+L_{cls}+L_{seg},$$

The semantic segmentation loss function $L_{cls}$ uses the BCE Loss, defined by the following formula:(5)$$L_{seg}=-\frac{1}{N}\sum_{m}^{N}\left[y_{m}\ln{\hat{y}}_{m}-\left(1-y_{m}\right)\ln\left(1-{\hat{y}}_{m}\right)\right],$$

In the equation above, *N* denotes the count of pixels, which is configured as 2 in our specific scenario (pertaining to disease leaf region and non-disease leaf region). The index *m* signifies the “*m*-th” pixel, ${\hat{y}}_{m}$ symbolizes the predicted class assigned by the algorithm to the “*m*-th” pixel, and $y_{m}$ corresponds to the ground truth (GT) label attributed to the “*m*-th” pixel.

## 3. Experiments

### 3.1. Datasets

To replicate scenarios with limited data, this study gathered images of diseased leaves and performed meticulous annotations. To mimic real-world detection conditions, the data collection was carried out using smartphone cameras in authentic planting environments. This encompassed varying angles and distances for image capture, simulating diverse perspectives. Notably, the same disease category exhibited distinct scales across different photographs, contributing to the intricacy of detection. Additionally, data collection took place under diverse weather and lighting conditions, encompassing fluctuations in intensity and color temperature. To further intensify the challenges, occlusions between leaves and complex backgrounds consisting of structures, soil surfaces, and concrete grounds were introduced, as highlighted in Figure 11.

A total of 300 images were amassed to replicate a small-scale dataset. Figure 12 illustrates instances of target detection annotations achieved using the LabelImg tool, alongside semantic segmentation annotations produced via the Labelme tool for disease regions and diseased leaves.

In the course of our experiments, given the high prevalence of rust disease and the relatively infrequent occurrence of other diseases, distinct disease types exhibited dissimilar shapes and color characteristics at various stages. Thus, a reclassification of disease types was undertaken, categorizing them into rust (as shown in Figure 13a), other diseases (as shown in Figure 13b), and mid–late stage diseases (as shown in Figure 13c). With respect to semantic segmentation tasks, a dataset for semantic segmentation was also curated, differentiating segmentation types into two categories: leaf areas with diseases and leaf areas without diseases.

In total, 4914 instances of disease targets and 473 instances of diseased leaves were meticulously annotated. Subsequently, the dataset was partitioned into training and testing sets in an 8:2 ratio. Specifically, 240 images were randomly allocated for network training, while the remaining 60 images constituted the testing dataset. The training set contained 3651 annotations for disease targets, and the testing set encompassed 1263 annotations, as outlined in Table 2. Within the training and testing sets, the instances of diseased leaves were 397 and 76, respectively.

Given the limited volume of data available, there was a concern that the trained model might exhibit sub-optimal generalization capabilities. To bolster the network’s aptitude for generalization, this study employed data augmentation techniques to expand the training set within this small-sample dataset. The combination of OpenCV and imgaug was harnessed to implement diverse augmentation methods on the collected images. These methods encompassed the addition of mist, introduction of Gaussian noise, manipulation of hue and saturation, and application of blurring. The augmented outcomes are visually depicted in Figure 14. Following the augmentation process, the training set was expanded to encompass a total of 606 images, as outlined in Table 3. Moreover, the count of annotated disease target bounding boxes within the training set surged from 3651 to 14,342, as detailed in Table 4. The strategic application of data augmentation effectively augmented the pool of positive samples available for training the model. This served to mitigate the limitations imposed by the constrained dataset volume.

### 3.2. Evaluation Metrics

In this study, the evaluation of the proposed model’s performance was based on the mean Average Precision (mAP) metric. The primary evaluation criterion was mAP@0.5, where the accuracy of target predictions was determined by an intersection over union (IoU) threshold greater than 0.5. Additionally, the mAP@0.5:0.95 metric was introduced to offer a comprehensive assessment of localization accuracy. The model’s performance across different target scales was gauged by calculating average precision values for small (mAP-S), medium (mAP-M), and large (mAP-L) targets.

Beyond accuracy evaluation, this study delved into practical deployment aspects by analyzing inference speed (Frames Per Second, FPS), computation volume (GigaFLOPS, GFLOPS), and parameter count (Megabytes, MB). While mAP measured the detection algorithm’s precision, FPS and GFLOPS provided insights into the efficiency of detection, affecting hardware costs in deployment scenarios. Additionally, the parameter count served as an indicator of the model’s size, which can impact storage requirements and computational performance. Collectively, these metrics provided a holistic view of the model’s performance, accuracy, efficiency, and scalability.

## 4. Results and Analysis

This chapter presents the results of our ablation experiments to demonstrate the effectiveness of the proposed modules and compares their performance against other existing models. The diversity in disease types across various plants poses a challenge for conducting standardized algorithm assessments. To address this, our study compares the proposed model against general object detection algorithms commonly used in the field. Specifically, we evaluate our model against two-stage models, namely Faster R-CNN and Cascade R-CNN. Additionally, since our proposed model is a one-stage detector, we compare its performance against classical one-stage object detection models, including SSD, YOLOv3 [32], RetinaNet, and YOLOv6 [33].

In Section 4.1, we address the issue of training with high-resolution images, which can consume significant GPU memory. Two approaches are employed to mitigate this challenge:Utilizing a sliding window technique to divide high-resolution images into smaller 640 × 640 sub-images.Downsampling the resolution of training images to 640 × 640.In this study, we compare the outcomes of these two methods. The dataset generated using the sliding window approach is referred to as the “sliding window cropped dataset”, while the dataset produced through downsampling is termed the “non-sliding window dataset”. Table 5 provides an overview of the annotation types present in both datasets. The process of window sliding for image cropping is visually depicted in Figure 15, and the resulting cropped images are showcased in Figure 16. Notably, each cropped image contains at least one diseased object, thanks to the image segmentation process.

The comparative analysis was conducted on both the sliding window cropped dataset and the non-sliding window dataset. It is important to highlight that our proposed model accommodates semantic segmentation annotation input, a feature not available in other models. The compatibility with different types of annotation inputs is summarized in Table 6.

### 4.1. Ablation Studies

To validate the effectiveness of the proposed performance-enhancement model, this section employs the original YOLOv5-L as the baseline. The model is improved by replacing the backbone network with the IMB module, substituting the FPN with the IFPN module, and integrating the DH module to replace the coupled Detection Head. Additionally, for training and testing on the non-sliding window dataset, the Seg module is introduced along with corresponding loss functions to guide the optimization direction during training. Experimental results are presented on both the sliding window cropped dataset and the non-sliding window dataset, with detailed outcomes in Table 7 and Table 8. The experimental findings indicate that our approach effectively enhances network performance in the complex scenario of the disease dataset.

As depicted in Figure 17, the replacement of CSPdarknet53 from YOLOv5 with the IMB module yields notable improvements. On the non-sliding window dataset, there are enhancements of 1.3% in mAP@0.5 and 0.3% in mAP@0.5:0.95. On the sliding window dataset, these improvements increase to 3.2% in mAP@0.5 and 0.8% in mAP@0.5:0.95. Moreover, the improved model exhibits reductions in parameter count and computation volume compared to CSPdarknet53.

With the incorporation of IFPN, the model showcases mAP@0.5 and mAP@0.5:0.95 enhancements of 1.1% and 0.4%, respectively, on the non-sliding window dataset. The improvements are even more pronounced on the sliding window dataset. IFPN effectively aligns feature maps, providing adaptive weighting during fusion and enhancing feature selection flexibility, which ultimately improves localization accuracy.

After integrating the DH module, testing on the non-sliding window dataset results in mAP@0.5 and mAP@0.5:0.95 improvements of 0.6% and 0.3%, respectively, with even more significant enhancements on the sliding window dataset. DH facilitates the selection of various feature types across multi-scale output feature maps. The classification branch emphasizes texture features, while the localization branch favors edge contour features. The decoupling of the detection task enhances feature selection flexibility for each task, allowing distinct parameter spaces and broader optimization possibilities.

When training with the Seg module, testing on the non-sliding window dataset demonstrates significant performance gains, with mAP@0.5 and mAP@0.5:0.95 both increasing by 1.2%. The Seg module, along with its corresponding loss function, guides the parameter optimization direction of the backbone network during training, focusing attention on disease leaves and suppressing background noise.

The baseline model (IMB) has a computation volume of 94.6 GFLOPs and 24.95 million parameters. The introduction of IFPN, DH, and Seg into the IMB base model leads to a 20 GFLOPs increase in computation volume and a 9.75 MB increment in parameter count. This enhancement contributes to a 5% improvement in mAP@0.5:0.95 and a 2.7% improvement in mAP@0.5. Focusing solely on the disease detection task, the removal of the Semantic Segmentation Head reduces the computation volume by 11.3 GFLOPs and the parameter count by 0.33 million, further enhancing detection speed.

Notably, substantial enhancements are observed in various mAPs, with a particularly noteworthy improvement in mAP-S, which is beneficial for datasets containing many small targets in plant disease instances. Alongside the reduction in parameter count and computation volume, as illustrated in Figure 17, these outcomes emphasize the superior performance of the introduced modules and model in effectively detecting disease within the dataset’s realistic context.

### 4.2. Comparison with Different Object Detection Networks

#### 4.2.1. Testing on Sliding Window Dataset

Table 9 and Table 10 present a comparative analysis between the five previously mentioned models and the model introduced in this study. Notably, the SSD512 model failed to achieve convergence during dataset training, resulting in missing experimental outcomes. The proposed model exhibited superior performance across most metrics when compared to the other models. Particularly, it outperformed other two-stage object detection algorithms in terms of the mAP-S metric, highlighting its enhanced precision in detection.

Turning to Table 10, the proposed model also stands out in terms of detection recall, surpassing mainstream models. Furthermore, Figure 18 provides information about detection speed, showing that the proposed model significantly outperforms YOLOv6-M. This discrepancy is attributed to the additional computational resources consumed by YOLOv6-M’s post-processing steps following model execution, leading to lower AVGFPS.

Figure 18 additionally illustrates a comparison of computational volume and average detection speed, avoiding an “EfficientNet-like” scenario, where reduced GFLOPS corresponds to slower detection speed. Notably, each model’s default input resolution was retained. The RCNN series and Retinanet utilized an input resolution of 1333 × 800, SSD used 512, YOLOv3 employed 608, and the remaining models adopted 640. This observation underscores that the proposed model in this study, compared to most contrastive models, not only demonstrates superior speed but also higher accuracy. It is noteworthy that training with the Segmentation and removing it during testing leads to certain improvements in detection speed.

In conclusion, the proposed model in this study strikes a harmonious balance between detection speed and accuracy. Unlike YOLOv5-L, which also emphasizes balance, the proposed model achieves both greater accuracy and swifter speed. This emphasizes the practicality of the proposed detection model in real-world scenarios involving the detection of daylily diseases.

#### 4.2.2. Testing on Non-Sliding Window Dataset

Furthermore, a comparative analysis was conducted among the mentioned models using the non-sliding window dataset. It is important to note that the non-sliding window dataset includes both object detection annotations and semantic segmentation annotations. Some of the models listed in Table 6 utilize only object detection annotations for loss calculation, while the model proposed in this study simultaneously employs both types of annotations to calculate loss.

Table 11 and Table 12 provide the performance results of various models trained and evaluated on the non-sliding window dataset. Due to the downsizing of original images within the non-sliding window dataset, the pixel representation decreases, leading to a decrease in the overall mAP value, particularly affecting the mAP-S metric, which focuses on the accuracy of detecting small objects. Notably, the SSD512 and RetinaNet models struggle to achieve convergence under these downsized conditions.

When analyzing Table 11, it becomes evident that the proposed model in this study surpasses other mainstream models across most metrics. While YOLOv6-M performs well in detecting small and large objects, the proposed model’s superior overall precision and comprehensive performance overshadow this difference. In comparison to YOLOv5-L, the proposed model demonstrates improvements in all metrics. Similarly, Table 12 supports these conclusions, with mAR-L slightly trailing behind YOLOv6-M.

Taken together, these findings highlight the proposed model’s efficiency in accurately detecting small objects even in downsampled images. The detection of small objects is often susceptible to background interference. Therefore, these experimental results showcase the model’s enhanced capability to handle complex background-object relationships and foliage intricacies.

Based on the results presented above, it is clearly demonstrated that the model introduced in this study exhibits superior performance compared to mainstream models on both the sliding window dataset and the non-sliding window dataset. The model demonstrates an enhanced ability to extract information from small objects and effectively differentiate complex backgrounds from targets. Across both datasets, the achieved detection accuracy surpasses that of YOLOv6-M and shows comprehensive improvement over YOLOv5-L. Furthermore, the proposed model’s inference speed is 1.17 times faster than YOLOv5-L and 1.27 times faster than YOLOv6-M, highlighting the clear superiority of the model introduced in this study.

## 5. Discussion

In this study, a total of six models were evaluated through testing, and the results of inference were visually presented. Utilizing an input resolution of 640 × 640, the entire image was input into the pre-trained models for detection. The inference results are depicted in Figure 19. Upon observation, it becomes evident that the two-stage models perform well across various scales, particularly the Cascade R-CNN. However, the single-stage Faster R-CNN, equipped with only a detection module, lags behind the two-stage models in terms of detection rate. Among the single-stage YOLO series models, YOLOv3 demonstrates relatively poorer performance, exhibiting lower disease detection rates compared to both the two-stage models and other YOLO series models.

Building upon the foundation of YOLOv5-L, the model proposed in this study exhibits improvements. As illustrated in Figure 19d, for example, the proposed model successfully detects leaf diseases in the middle of the image, where YOLOv5-L falls short. Similar trends are evident in Figure 19e, where YOLOv5-L fails to detect diseases in the middle of the image and on the leaf surface. On the other hand, YOLOv6-M incorrectly classifies the disease type of the lesion on the bottom-left leaf. Across the other images, it is apparent that our proposed model achieves detection rates similar to YOLOv6-M. Notably, in the bottom-right leaf detection, YOLOv6-M achieves a higher detection rate but incorrectly identifies the reflective region between leaves as a disease target. Across other instances, it is evident that, compared to YOLOv6, our model still displays some limitations in detecting large objects. Future algorithmic improvements should focus on addressing the detection of large objects.

This analysis strongly suggests that the incorporation of the algorithm proposed in this study leads to enhanced detection rates and accuracy for daylily disease targets in complex scenarios. This implies that the algorithm possesses an improved capability to address the task of daylily disease detection in real-world settings.

## 6. Conclusions

When performing disease target detection on daylily leaf images captured in real-world environments, existing models encounter challenges in accurately identifying disease targets due to complex backgrounds, varying lighting conditions, and significant variations in the appearance and size of the same disease type. Moreover, these models often make mistakes by classifying background objects as disease targets. To tackle these issues, this study introduces an object detection model called DaylilyNet, which leverages multi-task learning to guide the optimization process. The main contributions of this study are as follows:

**(1) Enhanced Global Spatial Modeling:** DaylilyNet incorporates a separable self-attention mechanism based on global spatial features, improving the modeling of spatial boundaries between leaves, diseases, and backgrounds. This refinement enhances the accuracy of detecting small targets by providing better spatial context.

**(2) Improved FaPN (IFPN):** The proposed IFPN module employs adaptive feature weighting during fusion, placing greater emphasis on large target information. This helps counteract the decrease in accuracy for detecting large targets caused by the addition of separable self-attention. Consequently, the overall disease localization performance is enhanced.

**(3) Decoupled Head (DH):** The DH module is introduced, where the target localization branch independently extracts relevant features and filters out noise that is irrelevant to target positioning. This approach leads to improved target localization performance.

**(4) Semantic Segmentation Task:** DaylilyNet adds a Semantic Segmentation Task Head along with corresponding loss functions. This allows the model to focus its attention more precisely on disease-inflicted leaves, thus enhancing the detection accuracy of disease targets at various scales.

Experimental results conducted on actual daylily disease leaf datasets demonstrate that the proposed DaylilyNet model outperforms other mainstream object detection networks in terms of detection accuracy. It also surpasses most mainstream networks in detecting large, medium, and small targets. Additionally, the model showcases lower computational complexity and a reduced parameter count, which, in turn, leads to lower hardware requirements. It is important to note that the challenges faced in handling complex backgrounds are not unique to daylily diseases but are also relevant to the detection of other plant leaf diseases. Thus, the insights gained from this approach hold relevance for various leaf disease detection tasks.

However, it is important to acknowledge that the method proposed in this study requires the establishment of a semantic segmentation dataset. While manual annotation was previously relied upon, today, tools like Segment Anything (SAM) [34] simplify the annotation process. Nonetheless, while annotation tools have streamlined this process, they also raise the bar for dataset requirements. Consequently, the study intends to explore unsupervised or semi-supervised approaches in the future, such as few-shot and zero-shot learning, to further mitigate data requirements for model training.

## Figures and Tables

**Figure 1 sensors-23-07879-f001:**
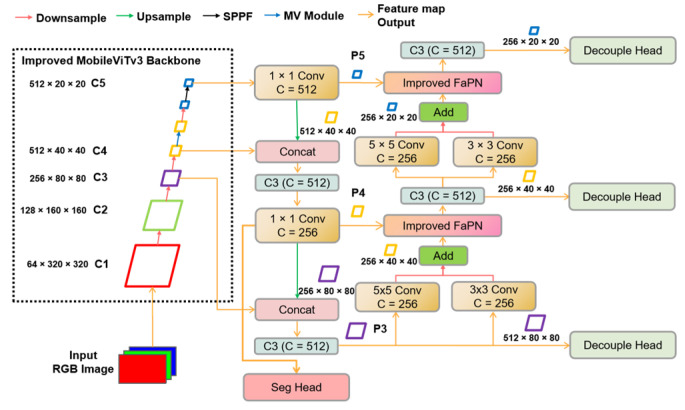
The general structure of the multi-task learning model proposed in this study.

**Figure 2 sensors-23-07879-f002:**
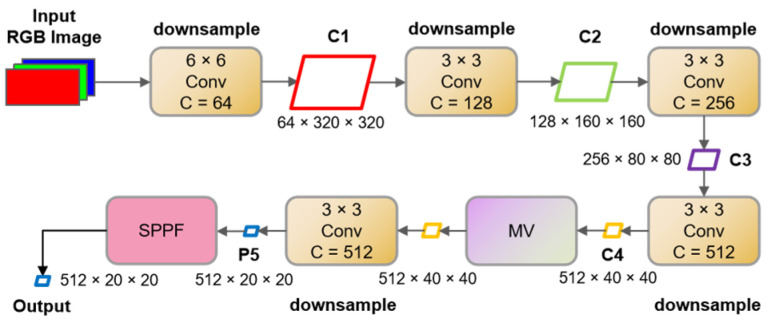
The structure of the IMB backbone.

**Figure 3 sensors-23-07879-f003:**
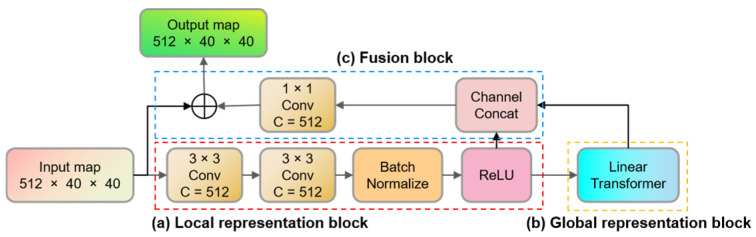
The structure of the enhanced MV module. It includes three submodules: (a) Local representation block, (b) Global representation block, and (c) Fusion block.

**Figure 4 sensors-23-07879-f004:**
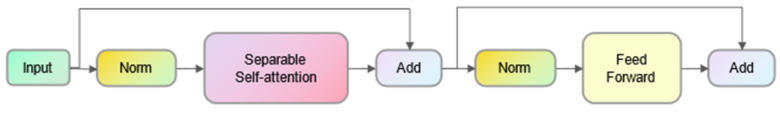
The structure of the linear transformer module.

**Figure 5 sensors-23-07879-f005:**
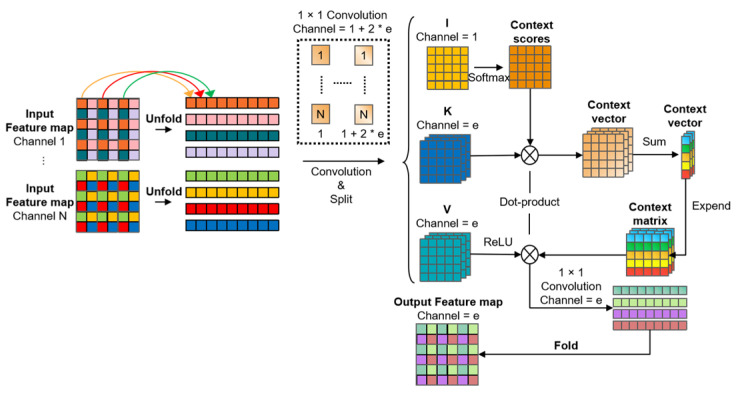
The structure of SSA.

**Figure 6 sensors-23-07879-f006:**
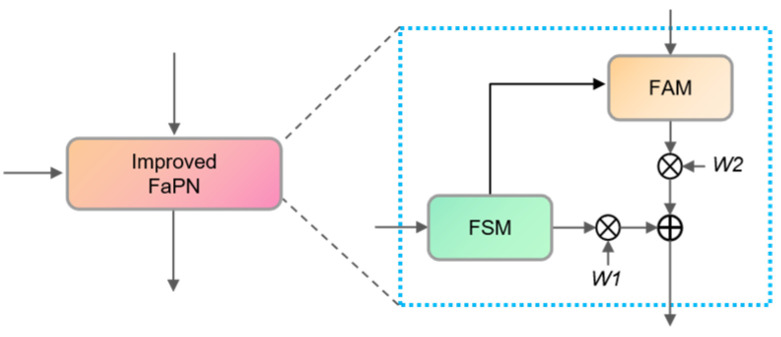
The structure of the IFPN.

**Figure 7 sensors-23-07879-f007:**
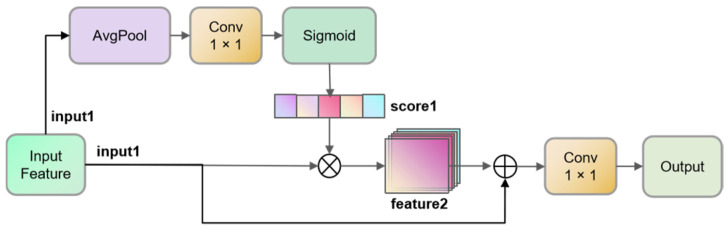
The structure of the FSM.

**Figure 8 sensors-23-07879-f008:**
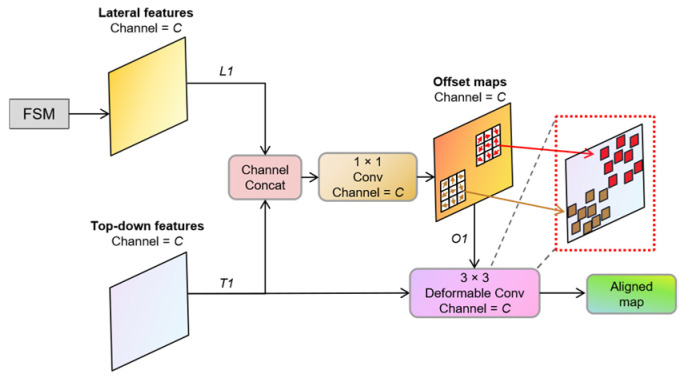
The structure of the FAM.

**Figure 9 sensors-23-07879-f009:**
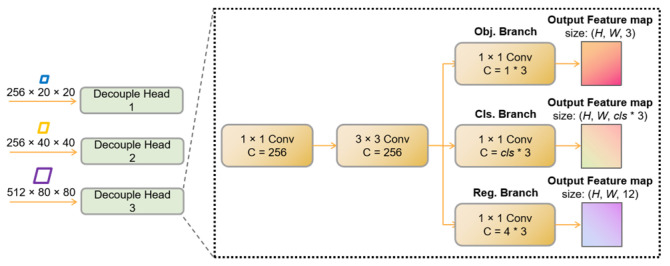
The structure of the Decoupled Head.

**Figure 10 sensors-23-07879-f010:**
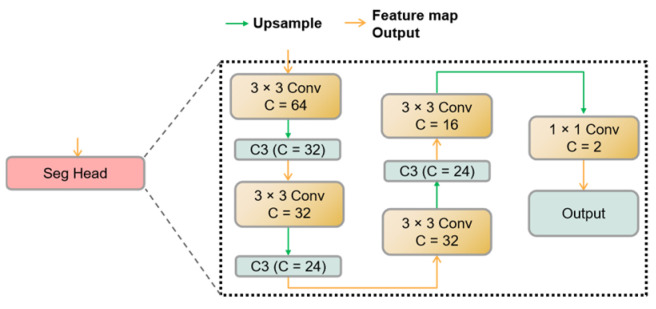
The structure of the Segmentation Head.

**Figure 11 sensors-23-07879-f011:**
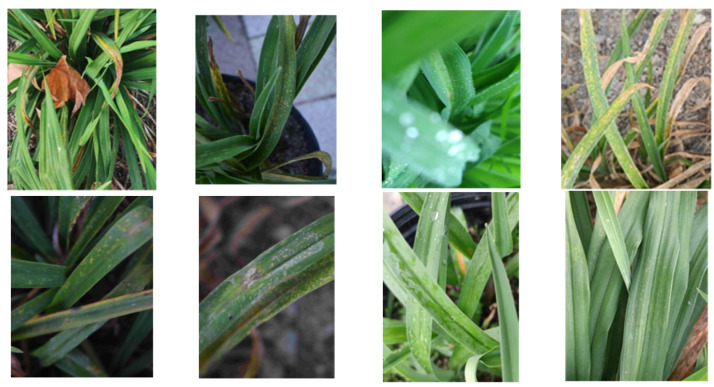
Pictures of data collected in the field.

**Figure 12 sensors-23-07879-f012:**
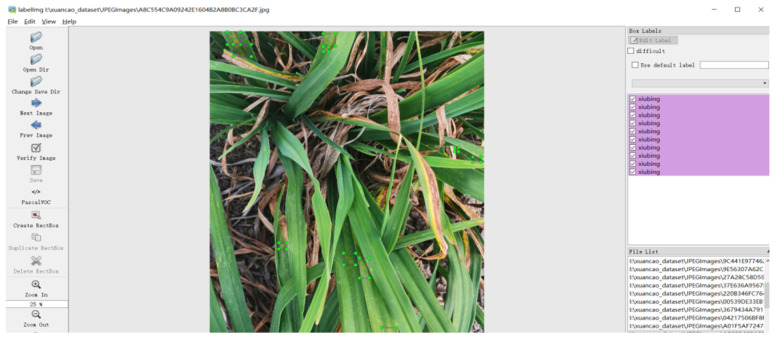
Interface of LabelImg software.

**Figure 13 sensors-23-07879-f013:**
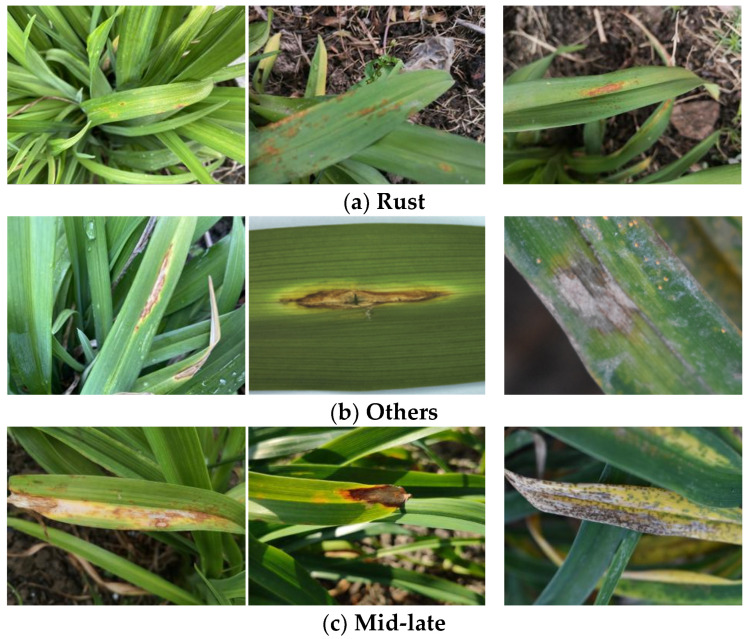
Characteristics of different diseases.

**Figure 14 sensors-23-07879-f014:**
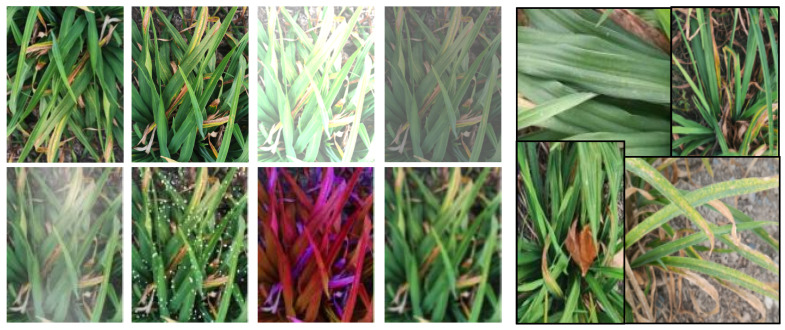
Images generated with the augmentation technique.

**Figure 15 sensors-23-07879-f015:**
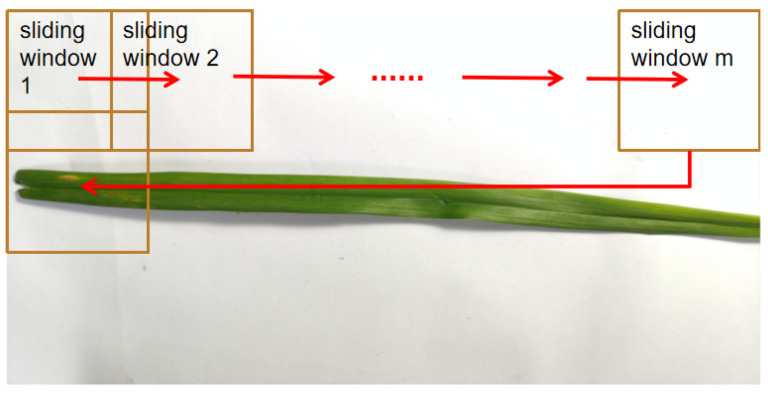
Schematic diagram of sliding window cutting algorithm.

**Figure 16 sensors-23-07879-f016:**
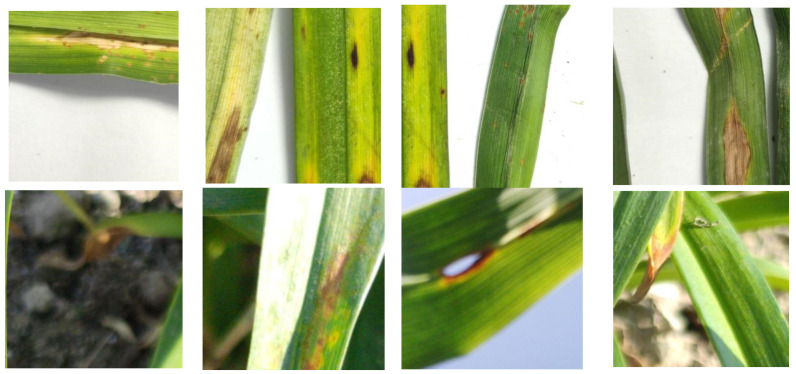
Images cut by sliding window cutting algorithm.

**Figure 17 sensors-23-07879-f017:**
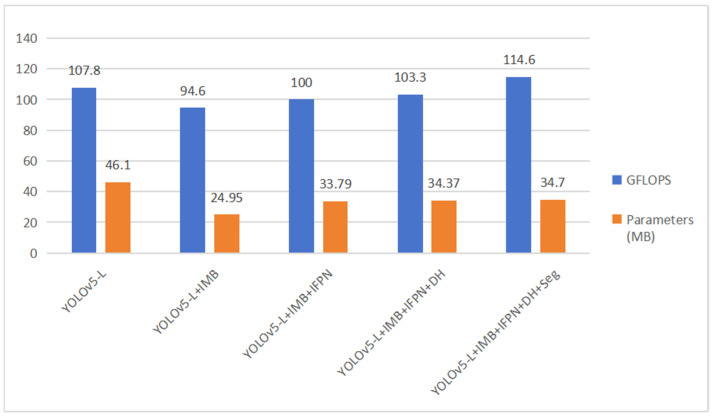
Impact of different modules on overall computation volume and parameter count.

**Figure 18 sensors-23-07879-f018:**
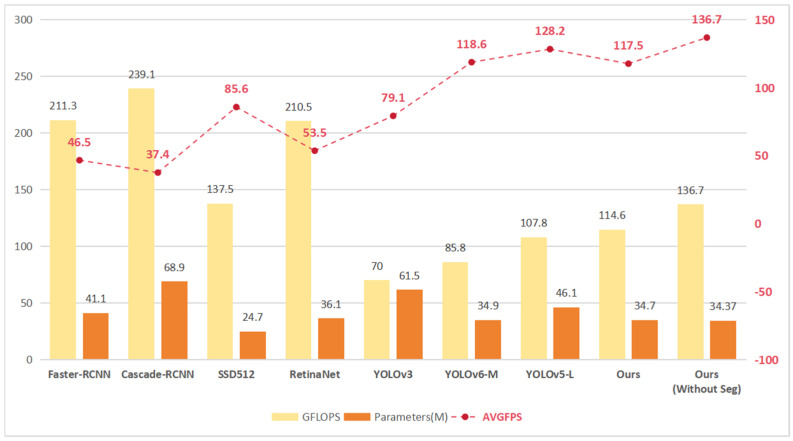
Comparison of 8 models in terms of number of parameters, computation and detection speed. Ours (Without Seg) has the Segmentation removed during the testing phase.

**Figure 19 sensors-23-07879-f019:**
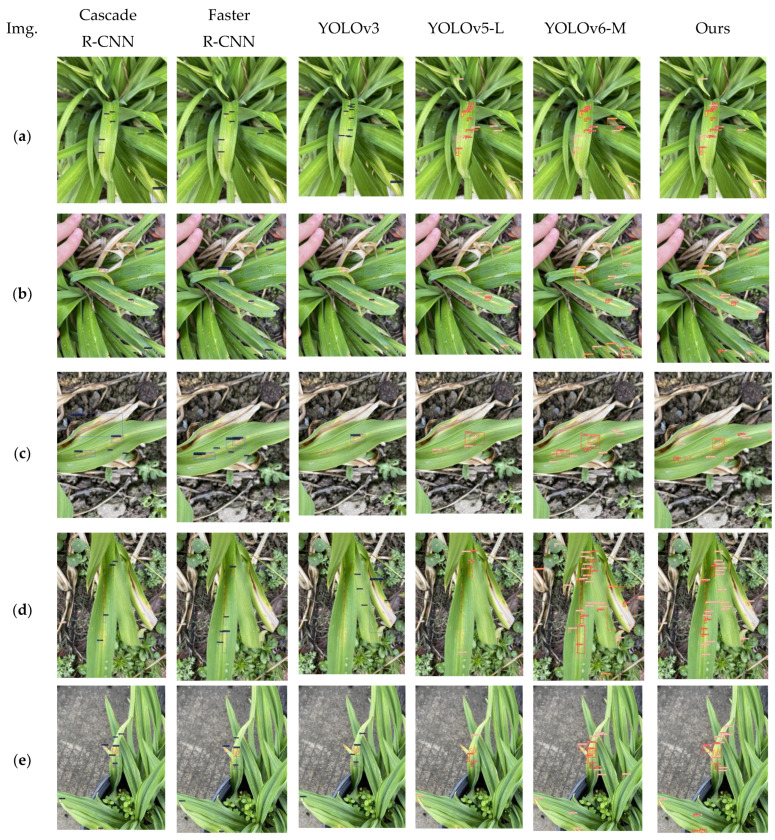
Comparison of experimental results of 6 models. There are a total of 5 images, labeled as (**a**) to (**e**) respectively. These images were captured in real-life scenarios, and each column displays the detection results of a model for images (**a**–**e**).

**Table 1 sensors-23-07879-t001:** The computational complexity of MHSA and SSA.

Methods	Complexities
Multi-head self-attention	*O*(*k*^2^)
Separable self-attention	*O*(*k*)

**Table 2 sensors-23-07879-t002:** Number of data and number of target box annotations.

Dataset Type	Rust	Others	Mid–Late	Total
Training set	1344	2153	154	3651
Validation set	456	767	40	1263

**Table 3 sensors-23-07879-t003:** The number of images in the dataset before and after data augmentation.

Data Augmentation	Image Count	
	Training Set	Validation Set
No augmentation	153	41
With augmentation	606	41

**Table 4 sensors-23-07879-t004:** The count of annotated targets in the dataset before and after data augmentation.

Annotation Type	Disease Type	Training Set Object Count		Validation Set Object Count
		* No Aug	* With Aug	
bbox	Rust	1344	5306	456
	Others	2153	8420	767
	Mid–late	154	616	40
-	Count	3651	14,342	1263

* No aug means without augmentation, With aug means with augmentation.

**Table 5 sensors-23-07879-t005:** An overview of the annotated data types included in different datasets.

Datasets	Detection Data	Segmentation Data
Sliding window dataset	Yes	No
Non-sliding window dataset	Yes	Yes

**Table 6 sensors-23-07879-t006:** The models’ support for different annotated data types.

Models	Support	Applicable Datasets
Faster R-CNN, Cascade R-CNN, SSD512, YOLOv3, Retinanet, YOLOv6-M, YOLOv5-L	* Det.	1. Non-sliding window dataset (for object detection) 2. Sliding window dataset
Ours	* Det. + * Seg./* Det.	1. Non-sliding window dataset 2. Sliding window dataset

* Det. stands for object detection data, and Seg. represents semantic segmentation-type data.

**Table 7 sensors-23-07879-t007:** Ablation experiments with sliding window dataset.

Models	mAP @0.5:0.95 (%)	mAP @0.5 (%)	mAP-S (%)	mAP-M (%)	mAP-L (%)
YOLOv5-L	22.6	44.1	17.6	25.2	27.3
YOLOv5-L + IMB	23.4	47.3	22.1	28.4	25.2
YOLOv5-L + IMB + IFPN	26.2	48.3	23.5	31.8	30.2
YOLOv5-L + IMB + IFPN + DH	26.5	49.3	23.8	29.1	31.7

**Table 8 sensors-23-07879-t008:** Ablation experiments with non-sliding window dataset.

Models	mAP @0.5:0.95 (%)	mAP @0.5 (%)	mAP-S (%)	mAP-M (%)	mAP-L (%)
YOLOv5-L	13.8	29.1	7.5	20.8	17.7
YOLOv5-L + IMB	14.1	30.2	7.9	21.1	17.7
YOLOv5-L + IMB + IFPN	14.5	31.3	8.5	21.4	17.1
YOLOv5-L + IMB + IFPN + DH	14.8	31.9	8.8	22.7	16.8
YOLOv5-L + IMB + IFPN + DH + Seg	16.0	33.1	9.9	24.3	19.1

**Table 9 sensors-23-07879-t009:** Comparison of detection results of 8 models on the sliding window dataset.

Models	mAP@0.5:0.95 (%)	mAP@0.5 (%)	mAP-S (%)	mAP-M (%)	mAP-L (%)
Faster R-CNN	17.0	31.5	14.2	4.9	27.0
Cascade R-CNN	17.1	32.4	16.0	5.1	25.4
SSD512	-	-	-	-	-
YOLOv3	12.4	28.7	10.1	5.3	21.9
Retinanet	13.2	24.8	11.3	4.0	22.0
YOLOv6-M	25.8	46.4	22.8	28.3	29.5
YOLOv5-L	22.6	44.1	17.6	25.2	27.3
Ours	26.5	49.3	23.8	29.1	31.7

**Table 10 sensors-23-07879-t010:** mAR results for detection results of 8 models on the sliding window dataset.

Models	mAR (%)	mAR-S (%)	mAR-M (%)	mAR-L (%)
Faster R-CNN	23.9	7.6	38.0	33.7
Cascade R-CNN	24.8	8.9	36.0	36.4
SSD512	-	-	-	-
YOLOv3	20.4	10.0	31.8	23.2
Retinanet	24.7	11.0	35.7	29.9
YOLOv6-M	57.8	49.2	53.5	60.8
YOLOv5-L	48.3	41.0	45.5	54.7
Ours	56.0	47.6	54.7	61.6

**Table 11 sensors-23-07879-t011:** Comparison of detection results of 8 models on the non-sliding window dataset.

Models	mAP@0.5:0.95 (%)	mAP@0.5 (%)	mAP-S (%)	mAP-M (%)	mAP-L (%)
Faster R-CNN	14.5	29.5	7.8	22.0	21.2
Cascade R-CNN	16.3	32.4	4.8	24.5	23.7
SSD512	-	-	-	-	-
YOLOv3	12.4	28.7	5.3	21.9	18.2
Retinanet	-	-	-	-	-
YOLOv6-M	14.6	29.8	10.8	20.0	20.7
YOLOv5-L	13.8	29.1	7.5	20.8	17.7
Ours	16.0	33.1	9.9	24.3	19.1

**Table 12 sensors-23-07879-t012:** mAR results of 8 models on the non-sliding window dataset.

Models	mAR (%)	mAR-S (%)	mAR-M (%)	mAR-L (%)
Faster R-CNN	23.9	7.6	38	33.7
Cascade R-CNN	24.8	8.9	36	36.4
SSD512	-	-	-	-
YOLOv3	20.4	10	31.8	23.2
Retinanet	-	-	-	-
YOLOv6-M	34.5	30.0	43.5	29.9
YOLOv5-L	27.7	30.5	41.5	27.8
Ours	35.2	46.6	44.6	25.3

## Data Availability

The research data can be found at: https://github.com/HelloSZS/DaylilyNet-Data (accessed on 5 September 2023).

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
