# Peer review of "DaylilyNet: A Multi-Task Learning Method for Daylily Leaf Disease Detection"

_sensors, 2023, doi:10.3390/s23187879_

Round 1

Reviewer 1 Report

The authors have clearly demonstrated that the model introduced in this study exhibits superior performance compared to mainstream models on both the sliding window dataset and the non-sliding window dataset. A total of six models were evaluated through testing, and the results of inference were visually presented. Utilizing an input resolution of 640x640, the entire image was input into the pre-trained models for detection. The proposed model successfully detects leaf diseases in the middle of the image, where YOLOv5-L falls short. Similar trends are evident in Figure 19(e), where YOLOv5-L fails to detect diseases in the middle of the image and on the leaf surface. Also, YOLOv6-M incorrectly classifies the disease type of the lesion on the bottom-left leaf. This is achieved by a feature alignment module that improves the localization accuracy by mitigating feature misalignment. The article nicely covers the leaf disease detection using the proposed model based on an Improved MobileViTv3 backbone. However, some explanations are still required:

1.    What is the significance of sliding window dataset and the non-sliding window dataset ? This can be put in the abstract as well.

2.    How does the semantic segmentation task head along with corresponding loss functions enhancing the detection accuracy of disease targets at various scales?

3.    What is the main process used for small object detection in the proposed DaylilyNet ? Is it simply the semantic segmentation task ? If yes, it need to be explained in a much clear manner as mentioned in sl no 2.

No comment

Author Response

Dear Reviewer,

We would like to express our sincere gratitude to you for your valuable feedback on our manuscript. We appreciate the time and effort spent on the thorough evaluation of our work. We have carefully considered all the comments and suggestions provided by the reviewers, and we have made the necessary revisions to address their concerns. Below, we provide a point-by-point response to each of the reviewers' comments. 

Comment 1: What is the significance of sliding window dataset and the non-sliding window dataset ? This can be put in the abstract as well.

Response 1: Among these, the sliding window dataset involves splitting images into 640x640 segments using a sliding window approach, preserving the original pixel resolution. When inputting these segments into the model for detection, they are fed in their original form. On the other hand, the non-sliding window dataset takes the original resolution images and downsamples them to 640x640, resulting in a loss of certain fine details. The model receives these input images with loss of information. By comparing the detection results of these two preprocessing methods, we assess the performance degradation of the model when dealing with input data that has undergone information loss. A horizontal comparison of the results from various models allows us to effectively evaluate each model's ability to handle input data with information loss.

---

Comment 2: How does the semantic segmentation task head along with corresponding loss functions enhancing the detection accuracy of disease targets at various scales?

Response 2: In the training process, we input an image into our proposed model, which performs forward inference to obtain results for both semantic segmentation and object detection. Subsequently, we enter the phase of loss function optimization, where the results of object detection and the original annotations from the dataset are jointly input into the object detection loss function, while the results of semantic segmentation and the original annotations are likewise input into the semantic segmentation loss function. Finally, we sum the object detection loss and semantic segmentation loss to obtain the total loss, followed by overall backpropagation for optimization. Typically, object detection functions only involve the object detection loss, but our work incorporates joint optimization with the semantic segmentation loss. In this regard, we draw inspiration from the multi-task learning approaches such as YOLOP, which generally emphasize the benefits of introducing semantic segmentation tasks and their corresponding loss functions to focus the model's attention on the targets. Building upon these foundations, we designed our semantic segmentation task to align with the specific characteristics of our research.

---

Comment 3: What is the main process used for small object detection in the proposed DaylilyNet ? Is it simply the semantic segmentation task ? If yes, it need to be explained in a much clear manner as mentioned in sl no 2.

Response 3: In DaylilyNet, we have designed three separate object detection task heads for objects of three different scales (large, medium, and small). The inference process for detecting small objects follows the sequence: 'Image -> Backbone Network -> Small Object Detection Task Head.' On the other hand, the semantic segmentation task plays a role in adjusting the backbone network during the training process. This adjustment directs the backbone network's attention towards the leaves. Subsequently, during small object detection, the model utilizes the features related to the attention positions provided by the backbone network for further detection. It's important to note that the semantic segmentation task does not directly impact the small object detection during the inference process.

---

We thank you for your time and consideration. We look forward to your feedback and hope for a positive outcome regarding the publication of our work.

Sincerely,

Authorship team

Reviewer 2 Report

This paper proposes DaylilyNet, an object detection algorithm that uses multi-task learning for daylily leaf disease detection. By incorporating a semantic segmentation loss function, the model focuses its attention on diseased leaf regions, while a spatial global feature extractor enhances interactions between leaf and background areas. 

It is a valuable study. The manuscript is well-organized and well-written. References are sufficient and appropriate.

The followings can be fixed:

1- What are the running times (execution times) of the methods?

Additional results (a new table or chart/graph) may also be given in terms of running times.

2- The "Related Work" section is missing. A new section may be added between "Introduction" and "Section 2".  

Some parts of the "Introduction" section can be moved to the new section.  

Providing a table that summarizes the related work would increase the understandability of the difference from the previous studies in the "Related Works" section. 

3- There is only one sentence related to Figure 7. It can be explained in more detail. 

4- The organization of the paper (the structure of the manuscript) may be written at the end of the "Introduction" section. 

For example: "Section 2 presents ... Section 3 gives ...." 

5- Some abbreviations are used in the text without giving their expansion.   

For example; DETR, SOF-DETR, UNet, SSA, Relu, etc. 

The authors should write that "these abbreviations stand for what".

6- A concern is that no formal statistical analysis of the results are done, to indicate whether the differences in performance are statistically significant or not.

For example; Friedman Aligned Rank Test, Wilcoxon Test, Quade Test, etc.  

p-value can be calculated and compared with the significance level (p-value < 0.05). 

7- The symbols in the text should be italic. 

For example: 

- "applied to the feature map I along"

- "The feature map is divided into I, K"

- "For a feature map with a resolution of h * w"

- "the data shape from (b, c, h, w)"

- "The variable d corresponds to the token dimension"

- "predicting the coordinates (x, y) and dimensions (h, w) of the bounding boxes"

-

Author Response

Dear Reviewer,

We would like to express our sincere gratitude to you for your valuable feedback on our manuscript. We appreciate the time and effort spent on the thorough evaluation of our work. We have carefully considered all the comments and suggestions provided by the reviewers, and we have made the necessary revisions to address their concerns. Below, we provide a point-by-point response to each of the reviewers' comments. 

Comment 1: What are the running times (execution times) of the methods?

Additional results (a new table or chart/graph) may also be given in terms of running times.

Response 1: Due to the excessive number of tables in the paper, during the manuscript writing process, we contemplated presenting them using images. In Figure 18, we have indicated the 'red line,' which represents AVGFPS, equivalent to runtime. A higher AVGFPS corresponds to shorter runtime.

---

Comment 2: The "Related Work" section is missing. A new section may be added between "Introduction" and "Section 2".  

Some parts of the "Introduction" section can be moved to the new section.  

Providing a table that summarizes the related work would increase the understandability of the difference from the previous studies in the "Related Works" section. 

Response 2:

  • About the structure of Introduction

Since most papers in the reference journals adhere to the format of combining 'Related works' with the 'introduction,' we followed this convention during the writing process.

  • About the understandability

Due to the lack of research on 'Daylily Disease Detection,' it was challenging to compile this aspect into a table. Additionally, for the sake of improving understandability, when conducting research and citing references in the related work section, we listed our areas of focus (such as 'The existing plant disease detection based on deep learning method,' 'small object detection,' and 'multi-task learning') as headings and provided relevant information under each heading.

---

Comment 3: There is only one sentence related to Figure 7. It can be explained in more detail.

Response 3: At the time of writing, due to the need to control the amount of content, we simplified the descriptions of Figures 7 and 8. Following your advice, we have now added relevant descriptions.

---

Comment 4: The organization of the paper (the structure of the manuscript) may be written at the end of the "Introduction" section.

For example: "Section 2 presents ... Section 3 gives ...."

Response 4: Thank you for providing this feedback. We believe it is helpful for readers to understand the paper's structure, and we have added relevant content accordingly.

---

Comment 5: Some abbreviations are used in the text without giving their expansion.   

For example; DETR, SOF-DETR, UNet, SSA, Relu, etc.

The authors should write that "these abbreviations stand for what".

Response 5: Thank you for your thorough review. We have provided expansions for some of the abbreviations. In cases where these abbreviations were originally named in their shortened forms by their respective authors, we were unable to provide expansions.

---

Comment 6: A concern is that no formal statistical analysis of the results are done, to indicate whether the differences in performance are statistically significant or not.

For example; Friedman Aligned Rank Test, Wilcoxon Test, Quade Test, etc.  

p-value can be calculated and compared with the significance level (p-value < 0.05).

Response 6: Thank you for raising this question. One point I would like to emphasize is that when recording the results of our experiments, each model underwent multiple training runs. We excluded the best and worst two results and took the average of the remaining results. Therefore, when comparing these results with other models, they are indeed valuable for reference.

---

Comment 7: The symbols in the text should be italic.

For example:

- "applied to the feature map I along"

- "The feature map is divided into I, K"

- "For a feature map with a resolution of h * w"

- "the data shape from (b, c, h, w)"

- "The variable d corresponds to the token dimension"

- "predicting the coordinates (x, y) and dimensions (h, w) of the bounding boxes"

Response 7: Thank you for the detailed review. We have checked other sections and made corrections.

We thank you for your time and consideration. We look forward to your feedback and hope for a positive outcome regarding the publication of our work.

Sincerely,

Authorship team